# Environmental Factors Indirectly Impact the Nematode Carbon Budget of Subalpine Spruce Forests

Hongyang Zhou [1,2], Kaiwen Pan [1,*], Xiaoming Sun [1], Belayneh Azene [1,2], Piotr Gruba [3], Xiaogang Wu [1], Lin Zhang [1], Meng Zhang [1,2], Tianwen Tang [1,2] and Renhuan Zhu [1,2]

[1] CAS Key Laboratory of Mountain Ecological Restoration and Bioresource Utilization and Ecological Restoration and Biodiversity Conservation Key Laboratory of Sichuan Province, Chengdu Institute of Biology, Chinese Academy of Sciences, Chengdu 610041, China; zhouhy@cib.ac.cn (H.Z.); sunxm@cib.ac.cn (X.S.); belaynehazene@gmail.com (B.A.); wuxg@cib.ac.cn (X.W.); zhanglin@cib.ac.cn (L.Z.); zhaobo123789@gmail.com (M.Z.); renhuanzhu123@gmail.com (T.T.); huan_zr@163.com (R.Z.)

[2] University of Chinese Academy of Sciences, Beijing 100049, China

[3] Department of Forest Ecology and Silviculture, University of Agriculture in Krakow, Al. 29 Listopada 46, 31-425 Krakow, Poland; rlgruba@cyf-kr.edu.pl

[*] Correspondence: pankw@cib.ac.cn

**Abstract:** Nematodes play a significant role in soil biogeochemical cycling. However, our understanding of their community carbon budget response for a shift in the environmental conditions of natural and planted forests is limited. Therefore, we investigated the nematode community composition, daily carbon used in production and daily carbon budget, environmental variables, and the interaction among trophic groups in the moss, litter and 0–5 cm soil layers of natural subalpine spruce forest and plantations in western Sichuan, China. The result revealed that plantations increased the total nematode daily carbon budget by approximately 52% through the herbivore channel in the 0–5 cm soil layer. The herbivorous nematodes' daily carbon budget and production in the moss layer of plantations decreased by approximately 60% compared to natural forests. Nematode daily carbon used in production and carbon budget had a strong negative correlation with genus richness. The water content and total carbon was the most important environmental factor that affected the nematode carbon budget and production, respectively. However, the environmental factors indirectly affect the daily carbon budget of herbivore nematodes through omnivore top-down control in subalpine forest ecosystems. Our findings highlight that the planted ecosystems have a certain capacity to maintain abundance, richness, and carbon budget of soil nematode but increase the risk of herbivorous pests.

**Keywords:** nematode carbon budget; soil layers; soil food web; planted forest; environmental factors

## 1. Introduction

Planted forests (290 million ha) account for seven percent of the total forest area (4.06 billion ha) worldwide and increase by approximately three million ha per year, while the area of natural forests decreased by eight million ha per year, during 2010–2020 [1]. Globally, China has the most plantation forests (79.54 million ha) and accounts for approximately 36% of the total national forest area (220 million ha), largely due to the implementation of large-scale programs [1,2]. Thus, planted forests play an increasingly important role in carbon sequestration and biodiversity conservation [3]. It is generally acknowledged that the species richness and abundance across multiple trophic levels (i.e., vertebrates, invertebrates, and plants) in plantations is 32.7% and 15.7% lower than that in natural forests [4]. However, the diversity of soil fauna and microbes, as well as carbon stock density in plantations, was similar to natural forests [3,4]. Furthermore, most previous nematode studies focused on the top 15 cm of soil, with less attention being paid to the moss and litter layers in forest ecosystems [5,6]. Then, even though the moss and litter layers show more than twice the abundance of nematodes, with a predominance of predators

(mean occurrence 40%), compared with the soil layer [7,8]. Therefore, whether planted ecosystems are "green deserts" or suitable habitats for biodiversity, as well as the effects of biodiversity-related ecosystem services, remain unclear [4,9].

Nematodes are the most abundant multicellular organisms on earth and occupy a central position in the soil food web [10,11]. The nematode community regulates soil carbon (C) cycling by directly feeding the different forms of carbon in soil (i.e., microbial biomass carbon, old/recent soil carbon, root-derived carbon) to flow into the food web [12–14]. For example, the herbivore nematodes directly consume root productivity by approximately 4.4–20.2%, produce waste products and cadavers, shear root litter, and leak root exudates to induce C input to soil [15]. Bacterivore and fungivore nematode-selective grazing stimulated microbial-derived SOC retention or degradation through indirectly altering microbial community composition and their associated functional genes [12,16]. Omnivores and predators integrate the above herbivorous, fungivorous, and bacterivorous channels into a multichannel by trophic cascading control, which relates to both root- and microbial-derived SOC processes [10,17]. The nematode carbon budget is estimated based on the body size and abundance of each genus and obtained by the sum of the amounts that are respired and used for production [18,19]. It is sensitive to environmental changes and community assembly processes and indicates the magnitude of C flow into the soil food web [20]. Thus, the nematode carbon budget is meaningful to reflect their community characteristics and functions [20]. Furthermore, nematode individuals of different species occupy complementary ecological niches and enhance resource use, so that their carbon budget is enhanced by species richness and abundance [21]. These complementarity effects would be also offset or outweighed by interspecific and intraspecific competition [22]. Thus, a community with high nematode abundance or richness does not necessarily have a high carbon budget. However, our understanding of the nematode carbon budget response for shifts in the environmental condition in plantations and natural forests is limited.

Previous studies have shown that the soil environmental factors, including soil organic carbon (SOC), total nitrogen (TN), C/N ratio, and water content (WC), positively affect the nematode community's abundance, richness, and carbon budget across global or local levels [10,18]. It is generally accepted that plantation has negative impacts (8.4–36% decreased) on several soil properties, including SOC, TN, C/N ratio, and WC [23]. However, 18.1–42.5% of cases have reported that these soil properties were increased [23,24] or similar [3] in plantations compared to natural forests. Therefore, it is difficult to predict the effects of plantations on the nematode community's composition and function, as determined by these environmental factors. For example, previous studies have revealed that the plantations decreased the nematode abundance, taxa richness, and proportion of omnivores and predators by more than 30% compared to natural forests [25,26], while Kerfahi et al. [27] revealed that the omnivore and root parasites abundance increased by approximately 30–80%, but there was a similar richness in the plantation compared to natural forest. Furthermore, different nematode trophic groups interacted with each other through the soil food web relationship and indirectly changed the effect size of the environmental factors for nematodes, creating a more complicated relationship between the soil environment and nematode community. For example, in the nematode soil food web, the SOC indirectly inhibited predators and omnivores through interacting with bacterivores, so the total effect size of SOC on predator nematodes was weakened [28]. The environmental factors' effects are also enhanced by a positive interaction between different trophic groups [10]. Thus, the influence of those environmental factors on the nematode carbon budget needs to be reconsidered with food web interactions.

In this study, we aimed to explore the nematode carbon budget distribution pattern and response mechanisms for community composition, soil food web, and environmental condition shifts in natural and planted forests. We hypothesized that: (1) if nematode abundance or richness decrease, then their carbon budget and production would be decreased because higher abundance or richness enhance complementarity for resource use; (2) if the plantation environmental factors (i.e., SOC, TN, WC) decreased in plantation,

then the nematode carbon budget and production would be also decreased because poor environmental conditions lead to lower abundance and richness; (3) if the different trophic groups interact positively, then the effect size of the soil environmental factors for nematode carbon budget and production would be enhanced because environmental factors indirectly influence nematode carbon budget through the soil food web relationship.

## 2. Materials and Methods

### 2.1. Study Area

This study was conducted in the cold temperate coniferous forest zone at an altitude of 2800–3500 m above sea level in the Minshan Mountains, located in the transition belt of the eastern Tibetan plateau to Sichuan Basin, China (103.40–103.72 E, 32.24–33.10 N) (Figure 1A). The area is characterized by high mountains and deep valleys, and provides important ecological services for water and biodiversity conservation. Alpine and gorge ecosystems, along with primary forests, are the key conservation objects in this region [29]. The Minjiang and Baishuihe rivers (both branches of the Yangtze River) originate from this area. The study site has a plateau mountain climate, with a mean annual precipitation and temperature of 540.8–708.4 mm and 6.3–13.1 °C, respectively [30] (Figure 1E,F), typical brown forests soils, classified as Luvisols or Leptosols [31,32], and the greatest snow cover from late October to late March. The plant community of the tree layer is dominated by *Picea purpurea* Mast., *Picea asperata* Mast., and *Abies fargesii* Franch.; the shrub layer is dominated by *Ribes janczewskii* Pojark., *Lonicera similis* Hemsl., *Sorbus wilsoniana* Schneid., *Rosa omeiensis* Rolfe, and *Rubus sumatranus* Miq.; and the herb layer is predominantly *Thalictrum aquilegiifolium* Linnaeus, *Pilea notata* C. H. Wright, *Anaphalis sinica* Hance, *Geranium wilfordii* Maxim., and *Fragaria vesca* L.

### 2.2. Experimental Design and Sampling

The Songpan and Jiuzhaigou counties in western Sichuan province contain part of the degraded forest ecosystems in China. Major forest policies have been initiated since 1998, including the National Forest Conservation Program (NFCP), with the aim to restore natural forests, reforest arable areas with slopes greater than 25°, or regenerate both plantations and natural forests in the degraded areas for soil and water conservation [33]. In October 2018, five experimental areas (each containing both the spruce plantation and adjacent natural forest) were selected after field investigations (Figure 1A,D). The distance between adjacent experimental areas was more than 5 km. General information on the sampling areas is summarized in Table S1. In each experimental area in both plantation and adjacent natural forests, 20 m × 20 m plots were established in three replicates, with a distance of 30 m between adjacent plots (Figure 1D). Thus, a total of 30 plots (2 forest types × 5 experimental areas × 3 replicates) were established in this study (Figure 1A–D). Five sampling points (one at the center and one per corner of each plot) were selected. The moss, litter and 0–5 cm soil layers were sampled separately, and samples of the same layer collected from the five sampling points per plot were pooled and mixed thoroughly. The moss and litter layers were sampled from 10 cm × 10 cm monoliths. The moss layer sample included the closely associated organic matter that is important for moss nutrition [7]. Soil samples were collected from a depth of 0–5 cm, as this depth coincides with the highest abundance of most nematode species (including bacterivores, fungivores, and plant parasites) in woodlands [34]. All samples were divided into two parts, one of which was used for nematode extraction in a local sample site within two days of sampling and the other, for further analysis, was kept cool during transport.

### 2.3. Sample Analysis

The Baermann funnel method [35] was used to extract free-living nematodes from 100-g fresh moss, litter, and soil samples, collected from plantations and natural forests. The first 100 individuals of all the extracted nematodes were used for nematode identification at the genus level under a microscope (Upright Light Microscopes DM6B, Leica Microsystems,

Wetzlar, Germany), as described previously [36]. Nematode individual abundance was defined as the number of nematodes per 100 g of dry-weight of the sample. Based on their feeding characteristics, nematodes were allocated to five trophic groups: bacterivores, fungivores, herbivores, omnivores, and predators [37].

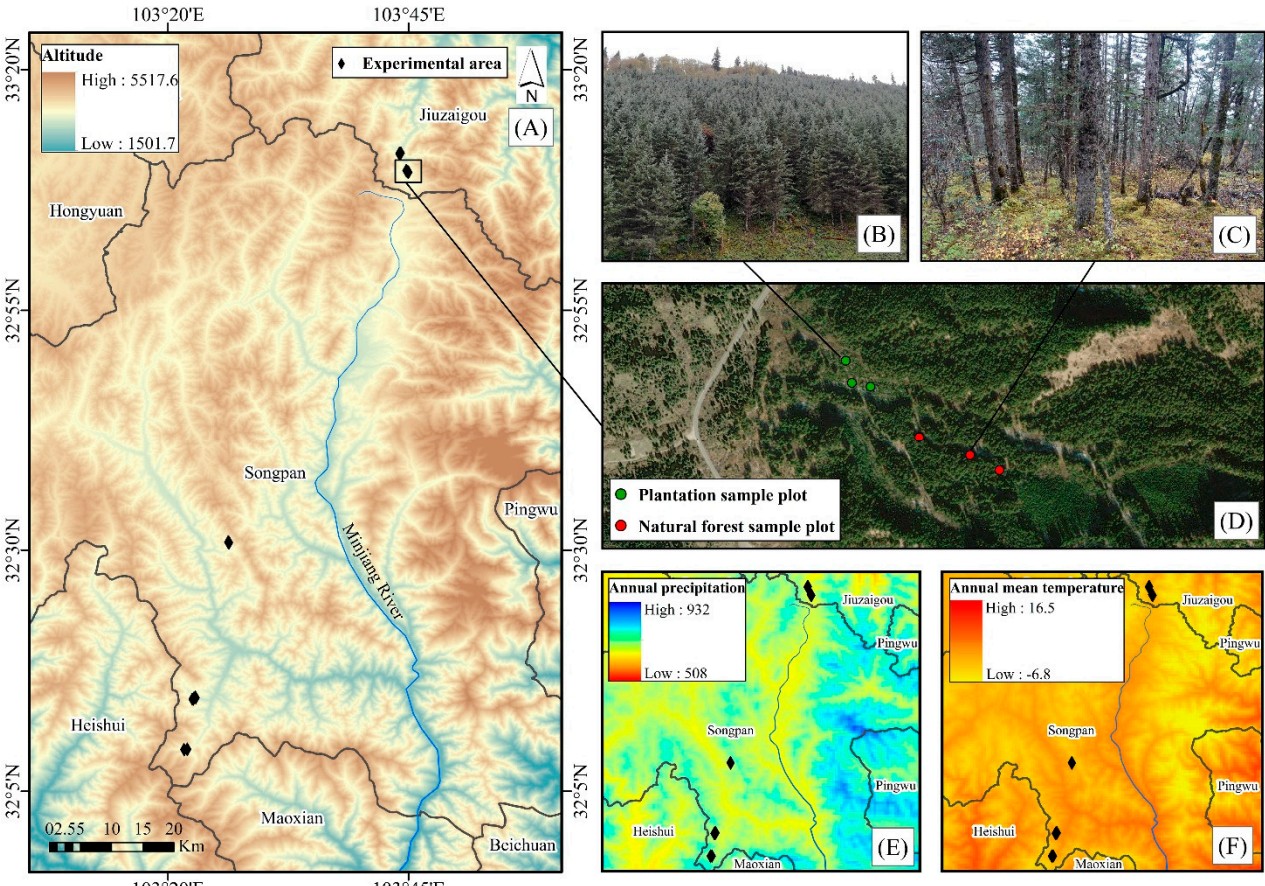

**Figure 1.** Location and general information of sampling sites in the Songpan and Jiuzhaigou counties in western Sichuan Province of China. (**A**): locations of the five experimental areas containing both the spruce plantation and adjacent natural forest; (**B**): plantation photograph; (**C**): natural forest photograph; (**D**): 20 m × 20 m sample plots with three replicates in plantation and natural forest; (**E**): diagram of annual precipitation; (**F**): diagram of annual mean temperature in the study area. The mean annual precipitation and temperature layer were found using the online resources https://worldclim.org/, accessed on 10 March 2022.

To calculate the nematode carbon budget and production, we first measured the nematode body size according to Zhao et al. [38]. Each identified nematode was photographed under a microscope, and its maximum body diameter (D) and body length (L) along the body curvature were measured using ImageJ [39]. In total, 7323 individuals were identified and measured. The fresh weight ($W_t$) of individual nematodes was calculated as $L \times D^2/1.6 \times 10^6$ at the genus level [18]. Individual nematode daily C used in production (DP) and daily C respiration were calculated as $0.104 \times W_t/12cp_t$ and $0.0159 \times W_t^{0.75}$, respectively, where $cp_t$ is the cp value of genus t [18]. The individual nematode daily carbon budget (DB) was obtained by the sum of the respiration part and the production part [18,20,40]. Finally, we multiplied the DP and DB value of individual nematode by individual abundance to compute the total DP and DB at the community level [18].

To assess the environmental factors, the thickness (T) and fresh weight (50 cm × 50 cm monolith, with five replicates) of the moss and litter layers were directly measured in each plot [41]. The biomass (B) of moss and litter were calculated using the water content and fresh weight. Water content was measured as the weight lost after oven-drying 20 g of

a moist field soil sample to a constant weight ($105 \pm 2$ °C) [42]. The moss layer water-holding capacity (WHC) was gravimetrically determined after immersing 100 g of fresh moss in water for 24 h. The bulk density (BD) of each soil sample was calculated based on its measured volume ($cm^3$) and dry weight. Soil pH was measured in a 1:2.5 soil: water suspension with a digital pH meter [42]. Soil inorganic N ($NO_3^-$-N, $NH_4^+$-N) was extracted from fresh soil samples using 2 M of KCl and determined using a flow injection autoanalyzer (AutoAnalyzer3, Bran + Luebbe, Norderstedt, Germany). Dissolved organic C and N (DOC and DON, respectively) contents of soil were extracted from 10-g fresh soil samples using 2 M of KCl. Soil microbial biomass C and N (MBC and MBN, respectively) contents were extracted by chloroform fumigation-$K_2SO_4$. The DOC, DON, MBC, and MBN were determined using a TOC/TN analyzer (Multi N/C®2100 (S), Analytik Jena AG, Jena, Germany). The total C and N contents (TC and TN, respectively) of all the layers were measured using an elemental analyzer (Vario MACRO cube, Elementar, Langenselbold, Germany), the C/N ratios were calculated as mass ratios [42]

### 2.4. Statistical Analysis

Statistical analyses were performed using SPSS (version 22.0 for Windows; IBM, Armonk, NY, USA). The data were first examined for the normality and homogeneity of variances. One-way analysis of variance (ANOVA) was used to test the effects of the three different layers (moss, litter, and soil) in plantations and natural forests on nematode DP and DB, as well as on environmental variables. The least significant difference (LSD) test was used for multiple comparisons. The independent-sample *t*-test was used to test differences among nematode DP and DB and environmental variables in the same layer of natural and plantation forests. Simple linear regression was used to analyze the relationships between nematode abundance and DP and DB. Statistical tests were considered significant at $p < 0.05$. The relationship between nematode C and environmental variables was analyzed by canonical correspondence analysis (CCA) using CANOCO (version 5.0 for Windows; Microcomputer power, Ithaca, NY, USA). Nonmetric multidimensional scaling (NMDS), based on Bray–Curtis similarity, was used to analyze the nematode community composition difference influenced by the forest types and substrate layers, using the vegan package for R [43]. Random forest was used to analyze the importance and significance of the environmental variables using the random Forest [44], A3 [45] and rfPermute (https://cran.r-project.org/web/packages/rfPermute/rfPermute.pdf, accessed on 10 March 2022) package for R [46]. Percentage increases in the mean squared error (MSE) of variables were used to estimate the importance of environmental variables [46].

Structural equation modeling (SEM) was carried out using AMOS (version 26 for Windows; IBM SPSS, Armonk, NY, USA) to analyze the interaction between the environment and the five trophic groups, with regard to soil depth and forest type, starting with an initial model (Figure S1) based on previous research on the soil food web [47]. Additional statistical texts ($\chi^2$, GFI and RMSEA) and their associated *p*-values were used to determine the fit of the SEM model. Before SEM, principal component analysis (PCA) was carried out to reduce the environmental variables, and four principal components (37.05%, 21.71%, 12.59%, and 11.70% of variance, explained by PC1, PC2, PC3 and PC4, respectively) were extracted to use to calculate the SEM model. Ten significant importance environmental variables (MB, LB, WHC, WC, TC, TN, C/N, $NH_4^+$-N, DOC, and pH) were put in the PCA according to the random forest analysis.

## 3. Results

### 3.1. Nematode Abundance and Richness in Natural and Planted Spruce Forests

A total of 116 genera were identified at the study site (Table S2). The results revealed that the nematode community composition was significantly influenced by substrate layers but not significantly by forest type (Figure 2). Plantation forests had a significantly higher individual abundance of total nematodes, herbivores, bacterivores, and genus richness of total nematodes than natural forest in 0–5 cm soil layers (58.39%, 55.98%, 73.72%, and

10.64%, respectively, $p < 0.05$) (Figure 3A,B,D,G). However, in the moss layer, the plantation forest had a significantly lower individual abundance of herbivores (52.08%) and genus richness of bacterivores (25.16%) than natural forest ($p < 0.05$) (Figure 3D,H). In addition, the individual abundance of total and trophic groups of nematodes in the 0–5 cm soil layer of both plantation and natural forests (except for herbivores in plantation forests) was significantly lower (28.5–97.11%) than in moss and litter layers ($p < 0.05$) (Figure 3A–F). The results also indicated that the moss layer had significantly lower total nematode genus richness in both forests and bacterivore genus richness in the plantation than in the litter (11.82–31.06%) and soil (8.78–38.18%) layers ($p < 0.05$) (Figure 3G,H). Similarly, in plantation forests, the genus richness of fungivores was significantly lower in the 0–5 cm soil layer than in moss and litter (approximately 36%) layers ($p < 0.05$) (Figure 3I). In both forests, herbivore genus richness was significantly higher in the 0–5 cm soil layer than in the moss (176.76% and 92.98% in plantation and natural forest, respectively) and litter (146.15% and 80.33% in plantation and natural forest, respectively) layers ($p < 0.05$) (Figure 3J). In the natural forest, the moss layer had significantly lower omnivore genus richness than the litter layer (37.93%, $p < 0.05$) (Figure 3K).

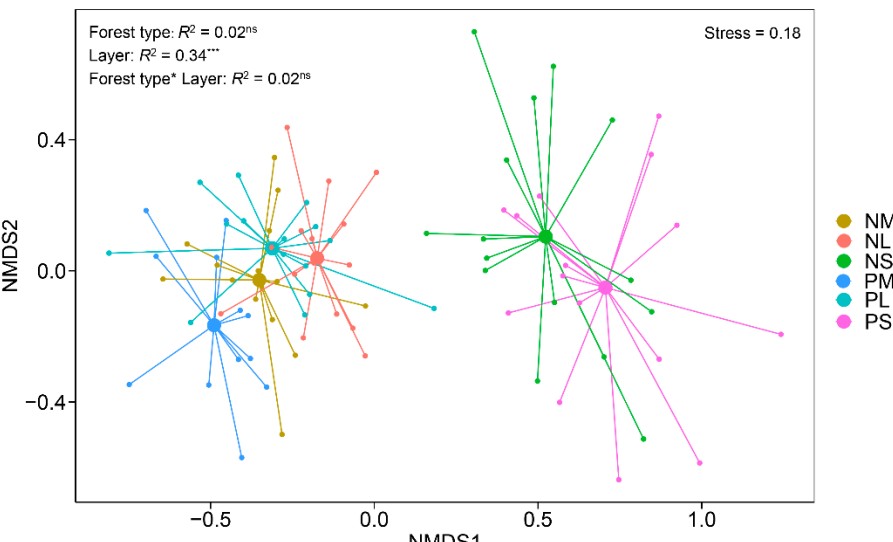

**Figure 2.** Non-metric multi-dimensional scaling (NMDS) based on Bray–Curtis similarity for the nematode community composition in subalpine spruce plantations and natural forests. PM: plantation moss layer; PL: plantation litter layer; PS: plantation soil layer; NM: natural forest moss layer; NL: natural forest litter layer; NS: natural forest soil layer. *: interaction of forest type and layer. ***: $p < 0.001$; ns: $p > 0.05$.

### 3.2. Nematode Daily Carbon Used in Production and Budget in Both Forest Types

The daily carbon used in production (DP) and daily carbon budget (DB) of the total, bacterivores, and predator nematodes in natural and planted forests were significantly decreased (74.99–97.95%) in the 0–5 cm soil layer compared to the moss and litter layers ($p < 0.05$) (Figure 4A,B,F–H,L). The results also revealed that the DB and DP of fungivores and omnivores in the litter layer were significantly higher (219.57–436.62%) than the soil layer in both forest types, while the DB of fungivores in the natural forest litter layer was also significantly higher (70.34%) than in the moss layer ($p < 0.05$) (Figure 4C,E,I,K). The DB and DP of herbivores were significantly lower in the moss layer than litter (50.34% and 48.38%, respectively) and soil layers (49.23% and 51.62%, respectively) of plantation forest ($p < 0.05$), but these variables showed no significant differences among the different layers in natural forests ($p > 0.05$) (Figure 4D,J). Compared with natural forests, the plantations showed a significantly higher DB of total nematodes (52.32%) and DP, DB of herbivores (59.07% and 61.05%, respectively) in the 0–5 cm soil layer ($p < 0.05$) but a significantly lower DP and DB of herbivores (60.24% and 60.27%, respectively) in the moss layer ($p < 0.05$)

(Figure 4A,D,J). However, no significant differences in other trophic groups were observed between the two forest types ($p > 0.05$) (Figure 4B,C,E–I,K,L).

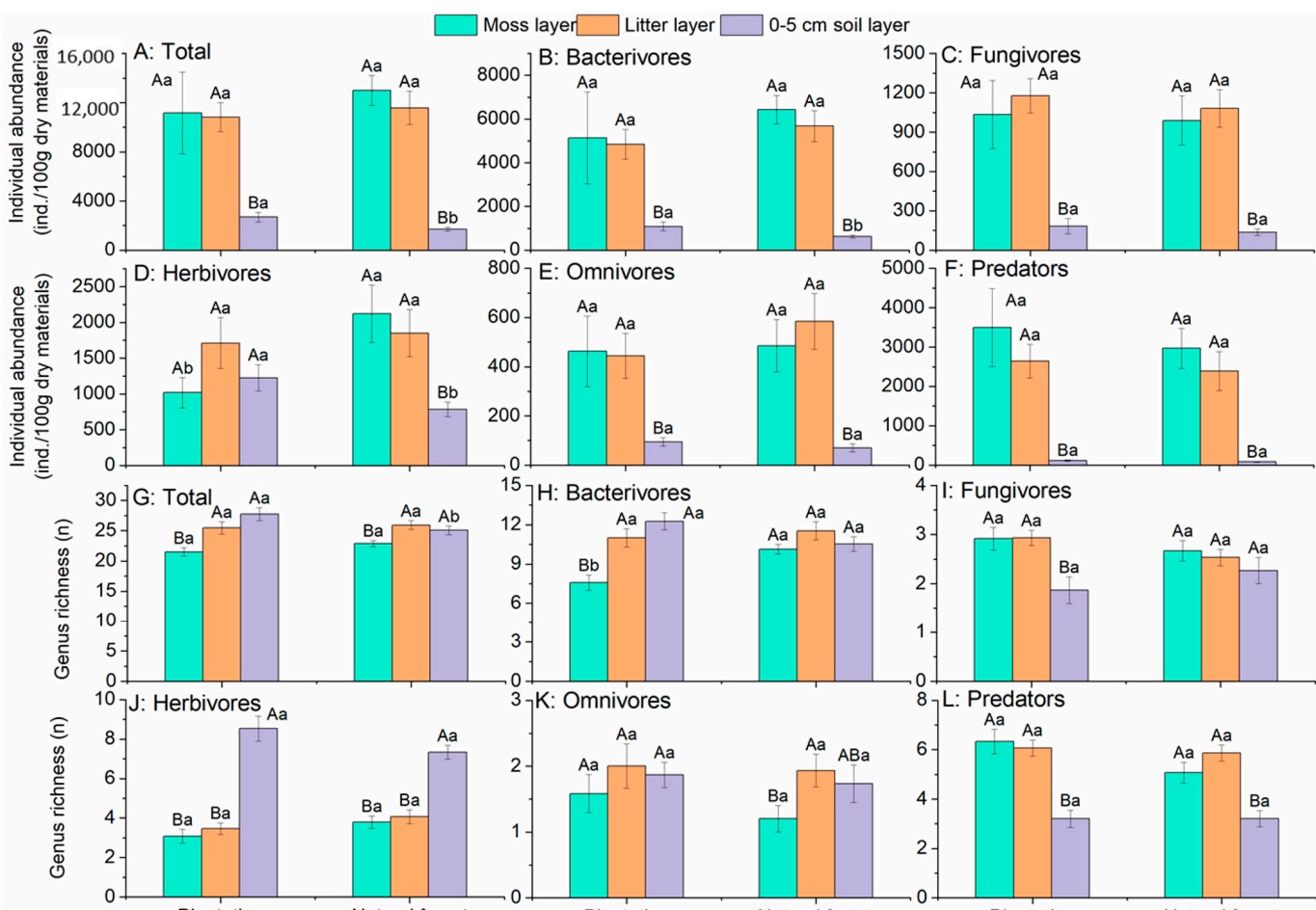

**Figure 3.** Total and trophic-group nematode individual abundance (**A–F**) and genus richness (**G–L**) in subalpine spruce plantations and natural forests. Upper-case letters indicate significant differences among the three layers (moss, litter, 0–5 cm soil); lower-case letters indicate significant differences between the two forest types.

### 3.3. Relationship of Nematode Community Composition and Nematode C

The nematode daily carbon used in production and daily carbon budget were significantly positively correlated with total individual abundance ($r^2 = 0.86$ and $r^2 = 0.87$; $p < 0.001$, respectively) (Figure 5A,C), while they were negatively correlated with total genus richness ($r^2 = 0.13$ and $r^2 = 0.12$; $p < 0.001$, respectively) (Figure 5B,D).

### 3.4. Effect of Environmental Factors on Nematode C

The WC, TC, and C/N ratio significantly decreased (34.03–82.96%) with increasing depth ($p < 0.05$), while TN was significantly lower in the soil layer than in the moss (52.91% and 63.31% in plantation and natural forest, respectively) and litter (52.99–64.45% in plantation and natural forest, respectively) layers in both forest types ($p < 0.05$) (Table 1). The WC, TC, TN, and pH of the soil layer were significantly higher (15.86–48.86%) in plantations than in natural forests ($p < 0.05$) (Table 1). However, the other environmental variables showed no significant differences between the two forest types or among the three substrate layers (Table 1).

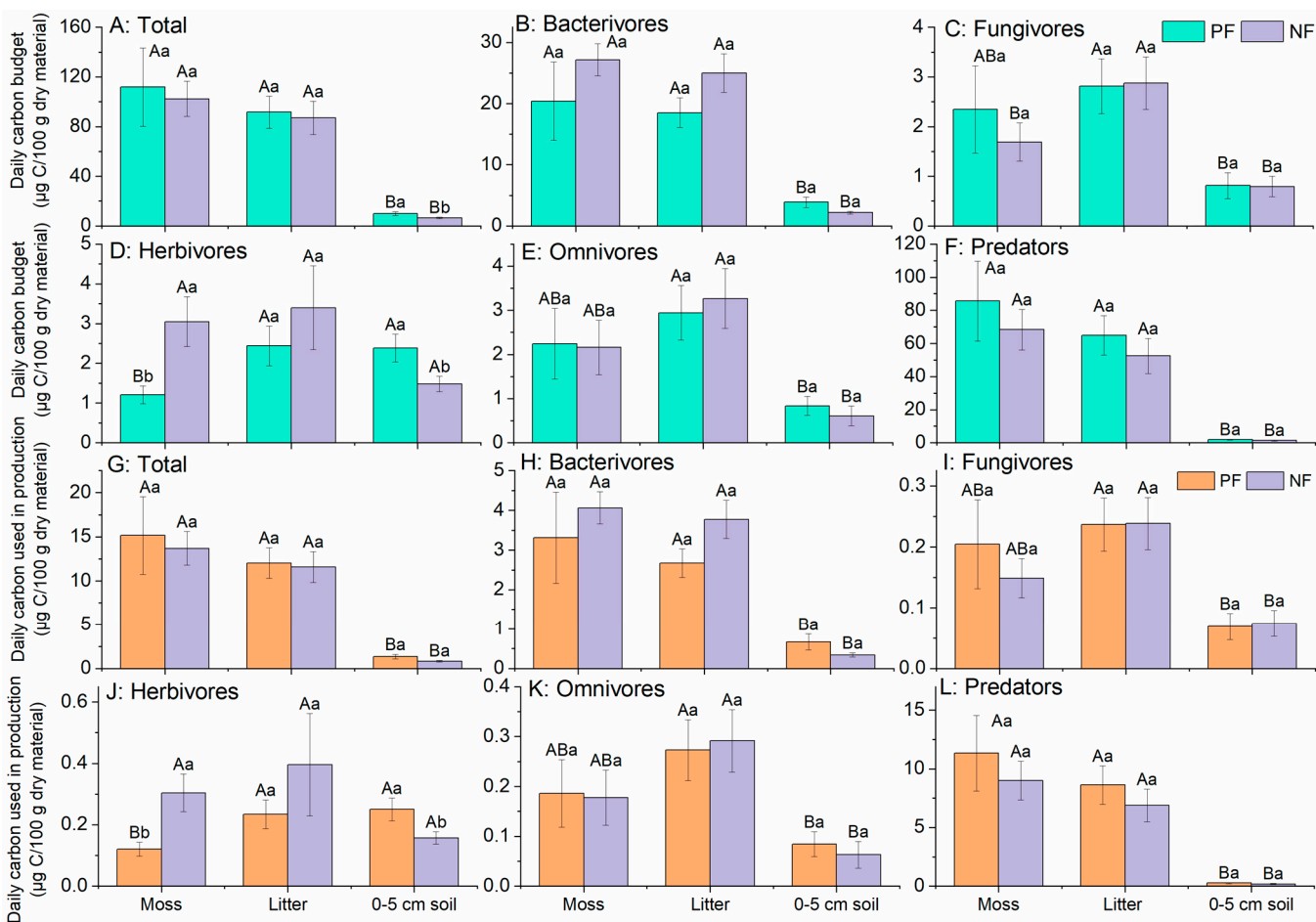

**Figure 4.** Daily carbon budget (**A–F**) and daily carbon used in production (**G–L**) of nematodes in total and different trophic groups in subalpine spruce forests. Upper-case letters indicate significant differences among the three layers; lower-case letters indicate significant differences between the two forest types.

The daily carbon used in production (DP) and daily carbon budget (DB) of total and predator nematodes was strongly correlated with environmental factors (Figure 6A). Random forest analysis showed that the WC, C/N ratio, TC, and TN were the most important environmental variables affecting total nematode DP and DB (Figure 4C,D). The biomass of litter and moss (LB and MB), and WHC was also important for omnivores (Figure 6B). Dissolved organic carbon (DOC) was the most important variable for DP and DB of fungivores (Figure 6B). Ammonium nitrogen content ($NH_4^+$) and pH were important for bacterivores (Figure 6B). However, the DP and DB of herbivore nematodes were poorly correlated with environmental factors (Figure 6A,B).

### 3.5. Effects of Soil Nematode Food Web on Nematode and Soil C

Environmental variables directly promoted the daily carbon used in production and daily carbon budget of bacterivores, fungivores, omnivores, and predator nematodes (Figure 7A–D). The omnivore daily carbon budget was directly facilitated by the fungivore bottom-up force (Figure 7B). The daily carbon budgets of herbivores and fungivores were directly facilitated by omnivore top-down force (Figure 7D).

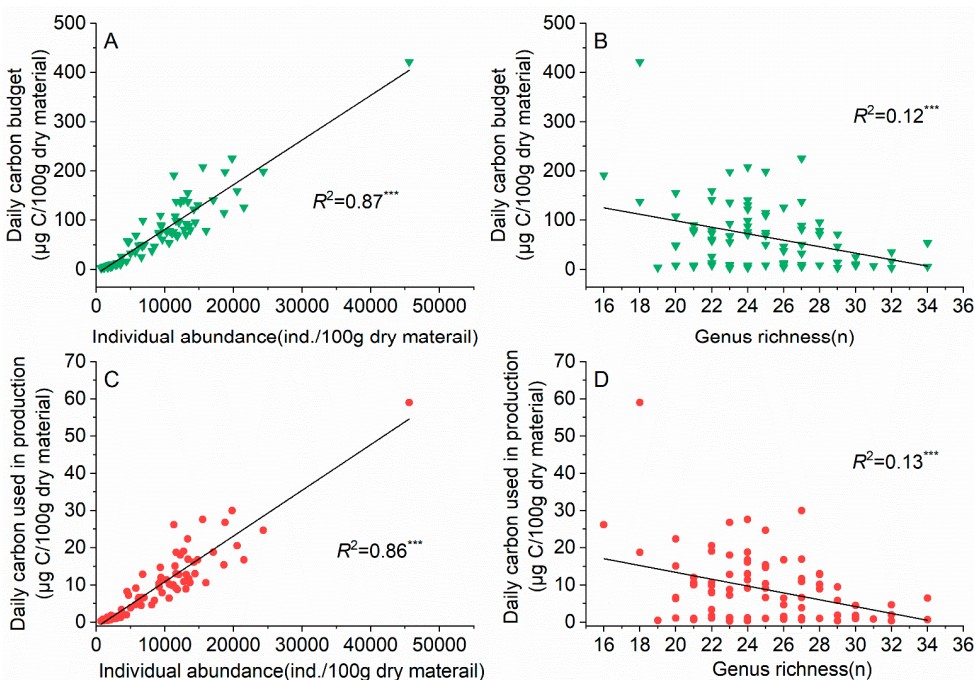

**Figure 5.** Linear regression analysis of daily carbon used in production and daily carbon budget to individual abundance and genus richness of nematodes in subalpine plantations and natural forests. (**A**): Daily carbon budget to individual abundance. (**B**): Daily carbon budget to genus richness. (**C**): Daily carbon used in production to individual abundance. (**D**): Daily carbon used in production to genus richness. *** $p < 0.001$.

**Table 1.** Environmental variables in subalpine spruce forests.

| Environmental Variable [1] | Forest Type [2] | | | | | |
|---|---|---|---|---|---|---|
| | **PM** | **PL** | **PS** | **NM** | **NL** | **NS** |
| WC (%) | 383.22 ± 9.72 aA | 287.01 ± 11.72 aB | 81.98 ± 6.46 aC | 373.72 ± 12.67 aA | 281.21 ± 10.88 aB | 63.68 ± 3.5 bC |
| TN (g/kg) | 17.37 ± 0.66 aA | 17.4 ± 0.5 aA | 8.18 ± 0.66 aB | 16.19 ± 0.34 aA | 16.71 ± 0.27 aA | 5.94 ± 0.41 bB |
| TC (g/kg) | 406.84 ± 6.78 aA | 362.05 ± 7.89 aB | 115.92 ± 12.9 aC | 406.98 ± 5.12 aA | 374.42 ± 10.01 aB | 77.87 ± 5.67 bC |
| C/N | 23.84 ± 1.12 aA | 20.92 ± 0.42 aB | 13.8 ± 0.6 aC | 25.29 ± 0.61 aA | 22.44 ± 0.62 aB | 13.18 ± 0.51 aC |
| T (cm) | 4.04 ± 0.44 a | 3.63 ± 0.48 a | - | 4.53 ± 0.5 a | 3.87 ± 0.46 a | - |
| B (g/m²) | 925.18 ± 84.19 a | 1041.94 ± 64.63 a | - | 1075.89 ± 110.78 a | 1304.44 ± 145.69 a | - |
| WHC (g/kg) | 6814.19 ± 338.99 a | - | - | 6524.17 ± 230.02 a | - | - |
| BD (g/cm³) | - | - | 0.58 ± 0.03 a | - | - | 0.65 ± 0.03 a |
| NO₃⁻ (mg/kg) | - | - | 18.05 ± 2.9 a | - | - | 11.73 ± 3.06 a |
| NH₄⁺ (mg/kg) | - | - | 8.67 ± 3.08 a | - | - | 7.49 ± 0.87 a |
| DOC (mg/kg) | - | - | 100.5 ± 11.45 a | - | - | 76.4 ± 14.97 a |
| DON (mg/kg) | - | - | 22.28 ± 3.93 a | - | - | 18.01 ± 2.23 a |
| MBC (mg/kg) | - | - | 587.03 ± 53.01 a | - | - | 511.33 ± 45.48 a |
| MBN (mg/kg) | - | - | 507.07 ± 62.6 a | - | - | 418.08 ± 48.46 a |
| pH | - | - | 5.26 ± 0.22 a | - | - | 4.54 ± 0.21 b |

[1] WC: water content; TN: total nitrogen; TC: total carbon; C/N: carbon to nitrogen ratio; T: thickness; B: biomass; WHC: water-holding capacity; BD: bulk density; NO₃⁻: nitrate nitrogen content; NH₄⁺: ammonium nitrogen content; DOC: dissolved organic carbon; DON: dissolved organic nitrogen; MBC: microbial biomass carbon; MBN: microbial biomass nitrogen; pH: soil pH. [2] PM: plantation moss layer; PL: plantation litter layer; PS: plantation soil layer; NM: natural forest moss layer; NL: natural forest litter layer; NS: natural forest soil layer. Uppercase letters indicate significant differences among the three soil layers; lowercase letters indicate significant differences between the two forest types.

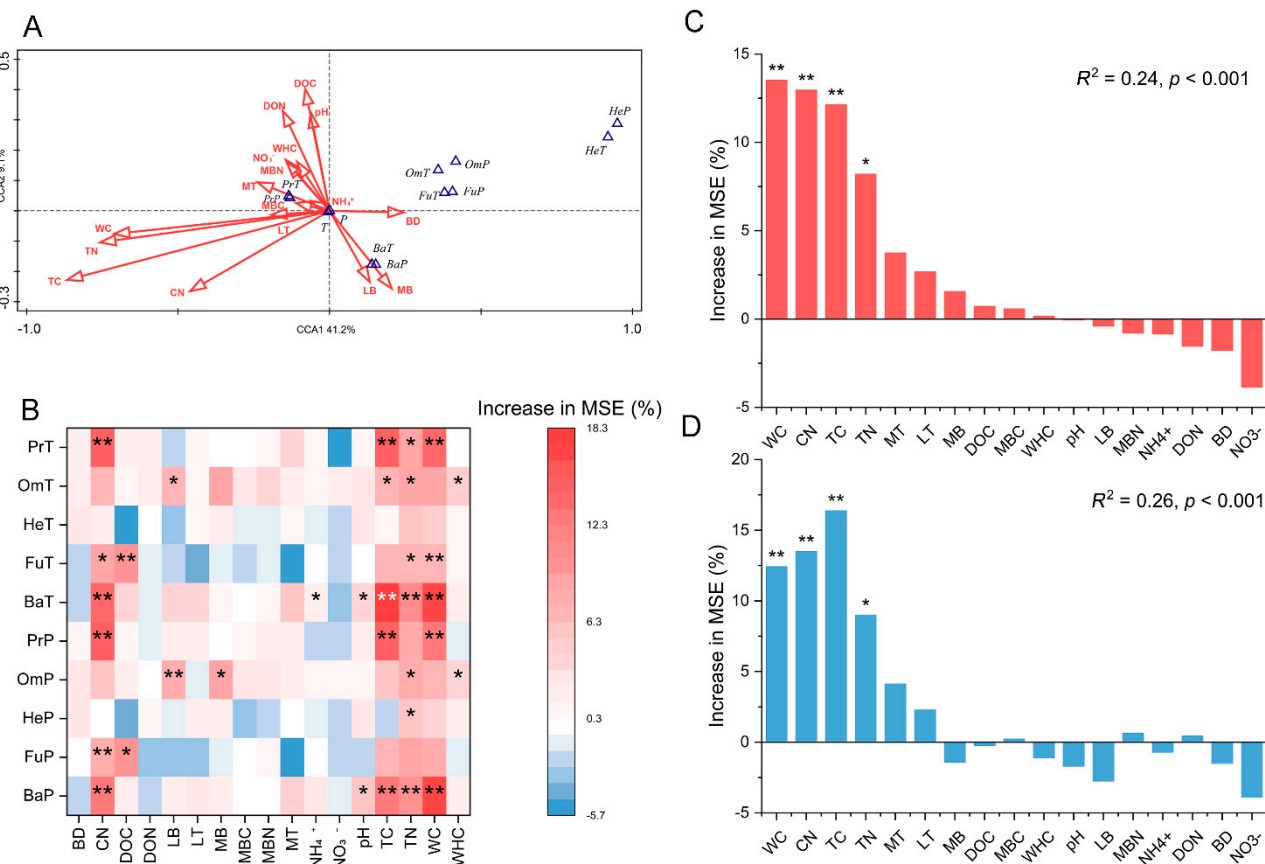

**Figure 6.** The relationship between environmental variables and nematode daily carbon budget and production. (**A**): Canonical correspondence analysis (CCA) of nematodes' daily carbon used in production and budget (total and in different trophic groups) and environmental variables in subalpine spruce forests. (**B**): Percentage increases in the mean squared error (MSE) (% of increase in MSE) and significance of environmental variables for daily carbon budget and production of different trophic group nematodes. (**C**): Percentage increases in the mean squared error (MSE) (% of increase in MSE) and significance of environmental variables for daily carbon budget of total nematodes. (**D**): Percentage increases in the mean squared error (MSE) (% of increase in MSE) and significance of environmental variables for daily carbon used in production of total nematodes. * $p < 0.05$, ** $p < 0.01$.

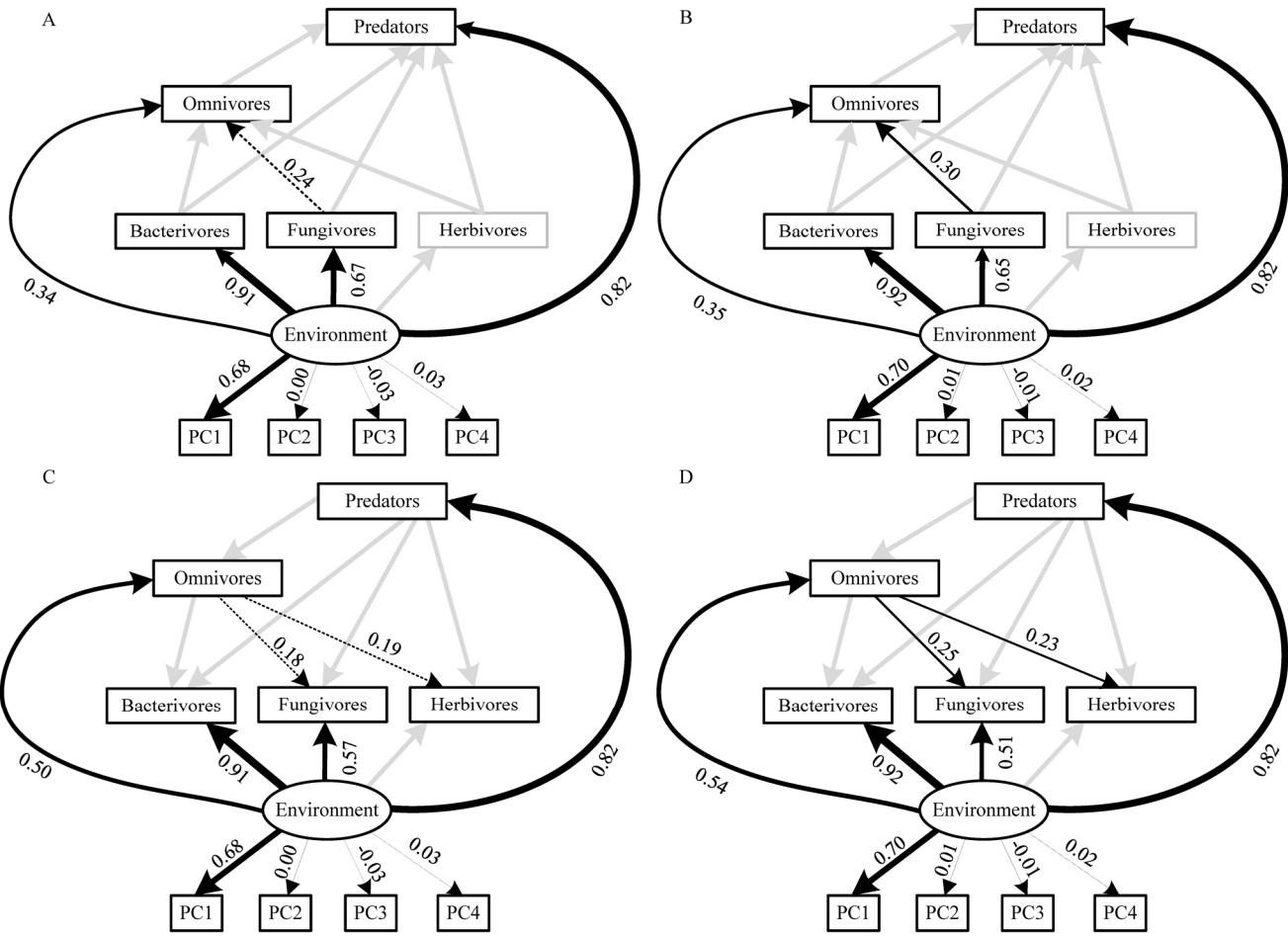

**Figure 7.** Structural equation modeling (SEM) of environmental variables and trophic group interactions of nematodes in subalpine spruce forests. (**A**) Bottom-up force for daily carbon used in production ($\chi^2$ = 20.912, *df* = 19, *p* = 0.342; GFI = 0.946, *p* = 0.499; RMSEA = 0.034, *p* = 0.579); (**B**) Bottom-up force for daily carbon budget ($\chi^2$ = 22.791, *df* = 19, *p* = 0.247; GFI = 0.941, *p* = 0.497; RMSEA = 0.048, *p* = 0.474); (**C**) Top-down force for daily carbon used in production ($\chi^2$ = 30.153, *df* = 26, *p* = 0.261; GFI = 0.933, *p* = 0.539; RMSEA = 0.043, *p* = 0.532); and (**D**) Top-down force for daily carbon budget ($\chi^2$ = 32.08, *df* = 26, *p* = 0.19; GFI = 0.929, *p* = 0.537; RMSEA = 0.052, *p* = 0.442). Numbers on arrows represent standardized path coefficients. The width of the arrow indicates the strength of the causal influence. A black solid arrow represents significant standardized path coefficient (*p* < 0.05), and a black dashed arrow indicates non-significant standardized path coefficient (*p* > 0.05). Grey arrows indicate paths removed to improve model fit.

## 4. Discussion

### 4.1. Nematode Genus Richness Negatively Correlated with Carbon Budget and Production in Subalpine Forest

In niche complementarity theory, the coexistence of different species in a community partitions their resource requirements, which finally facilitates the neighboring species' resource use of different foraging and life-history strategies [48]. A previous study showed that the increase in nematode abundance enhances the predation intensity, possibly inducing a higher C flow into the food web [12]. In the current study, the positive correlation between nematode individual abundance and C budget was also supported by the niche complementarity theory: increasing abundance enhances resource use so that the carbon budget would also be increased in plantation compared to natural forest [21]. This result is consistent with our first hypothesis.

However, the niche complementarity theory is not always supported by the previous and current studies. For example, Zhang et al. [20] indicated that the variation in genus

richness did not contribute to the increase in carbon budgets. Nielsen et al. [22] proposed that the relationship between soil species richness and ecosystem processes could be neutral. The neutral relation would be supported by the functional redundancy theory, which pointed out that the loss of a species does not change the ecosystem function, because different species in separate niches would organize the same function [49]. In functional redundancy theory, species numbers show an asymptotic response of soil processes, with maximal effects attained with few species [49]. Nielsen et al. [22] also proposed a 12% negative relationship between soil species richness and ecosystem processes when the species richness is greater than ten. In reality, the soil nematode community always comprises more than ten species [10,19,20], so the negative relationship was more possible in a real food web. In the current study, the nematode genus richness showed a strong negative correlation with daily carbon budget and production (Figure 5B,D), which is consistent with the study of Nielsen et al. [22]. With the increase in genus richness, the competitive interaction becomes stronger and beyond the effect of complementarity, which weakens the predation on carbon sources and finally decreases the daily carbon budget and production of the nematode community [22,50]. This result is inconsistent with our first hypothesis that a community with high nematode genus richness does not necessarily have a high carbon budget in subalpine spruce forests. Therefore, it is necessary to not only measure nematode community through abundance or richness, but also take the carbon budget into consideration.

### 4.2. Nematode Abundance, Richness, and Carbon Budget in Spruce Plantations and Natural Forests

Previous studies have found that natural forests had approximately 7–33% higher nematode abundance and taxa richness than plantations [25,26]. This could be attributed to natural forests having more soil fertility than plantations [23]. However, our findings were inconsistent with previous findings; the individual abundance and genus richness of total nematodes in the 0–5 cm soil layer in plantations were significantly higher (58.39% and 10.64%, respectively) than those in natural forests (Figure 3A,G). This could be attributed to the significantly higher values (15.86–48.86%) of environmental variables, such as soil WC, TN, TC, and pH, in plantations compared to natural forests (Table 1). A few studies also showed that SOC is higher in plantations than in natural forests (from 3% to 4%) [24], possibly because planted areas are well-managed, for example, by cutting tree branches to improve light conditions and tree health, which could have a positive long-term influence on the top-soil's physicochemical properties. In our plantation study areas, the sawdust residues left by logging would have increased the input of soil organic matter. These higher levels of environmental variables are likely to provide more abundant food resources for soil nematodes [51]. Consequently, the plantation significantly increased environmental conditions, such as water content, C and N in 0–5 cm soil, leading to a higher daily carbon budget (by approximately 52%) than natural forests (Figure 4A and Figure S3). However, the nematode carbon used in production was not influenced by forest type (Figure 4G). Previous studies reported that the carbon production of nematodes was significantly increased (62.7–147.1%) when residue mulching [19] or conservation tillage [52] were applied in agricultural ecosystems. This could be because agricultural systems have been subjected to more human disturbance than forest ecosystems. Therefore, the second hypothesis is partly consistent with our results. Plantations increase the nematode carbon budget but do not change the carbon used in production, which suggests that the nematode has reduced carbon use efficiency [19].

Yeates [51] reported higher (more than 81%) nematode abundance in the forest floor (moss, litter and humus) than in mineral soil. Similarly, high species richness and biomass were found in the moss layer, where larger nematodes such as *Aporcelaimellus obtusicaudatus* were abundant (6–22%) [7]. Powers et al. [8] reported that approximately 62% of the overall nematode genetic diversity existed in the litter and understory habitats, rather than in the soil. The current results revealed that the total individual abundance of nematodes in the moss and litter layers ranged from approximately four to eight-fold more than in the

0–5 cm soil layer of plantations and natural forests (Figure 3A) and approximately 95% of the nematode carbon budget and production in the moss and litter layers in both forests (Figure S2). These results suggest that ignoring the moss layer community would lead to a great underestimation of the nematode abundance and carbon budget in subalpine forests. However, previous studies paid less attention to comparing the effect of plantations on the moss layer nematode community. Our study highlights a simplified nematode community composition and a decreased herbivore carbon channel in the moss layer of the plantations and natural forests.

### 4.3. Environmental Factors Indirectly Promote Herbivore Carbon Budget in Subalpine Spruce Forest

A distinct daily carbon budget distribution pattern in plantations and natural forests was attributed to herbivores, which was the only trophic group affected by forest type. Herbivores feed on plant roots and epidermal cells as well as litter [37]. In plantations, the 5.13–16.13% higher moss layer coverage tends to homogenize the microhabitats, and the short needle litter of spruce easily pass through the moss layer into the litter and 0–5 cm soil layers, which provide more abundant food resources for herbivores. In natural forests, 20–60% higher herb coverage and more diverse plant species provide different forms of leaf litter, with the large leaf litter remaining in the moss layer and providing herbivores with a diverse food source (Table S1). However, the soil environmental valuables were not an important factor for the DB and DP of herbivores in this study (Figure 6A,B). This result was consistent with previous findings reported by Pan et al. [28], who found that environmental variables (represented by SOC) do not directly influence herbivores' metabolic footprint. Zhang et al. [10] also indicated that environmental variables (represented by SOC) have an indirect effect on herbivores through microbial biomass carbon. The effects that a notably different pathway of environmental variables have on herbivore nematodes demonstrated that the soil food web interaction obviously changes the effect that environmental factors have on the nematode community. In real webs, the herbivore channel constitutes the main food source for omnivores, which might be stimulated by a reduced predation intensity from omnivores [17]. The positive interaction between omnivores and herbivores enhanced the effect size of the soil environmental factors for nematodes in an indirect way. This finding was consistent with our third hypothesis. However, the herbivore nematodes, such as *Pratylenchus*, *Meloidogyne*, and *Xiphinema*, were considered as pests for trees [53], thus increasing the risk of nematode infection in plantations.

### 4.4. Reliability and Uncertainty

Our estimate for nematode biomass C for both the individual and community levels is within the range of previous studies [38,54]. These facts ensured that the carbon budget and production estimation were accurate. Our result also revealed that the nematode production C and the C budget in the 0–5 cm soil layer of plantations and natural forests are within the range of a temperate steppe [20]. Our estimate is, therefore, adequately reliable. However, our nematode production C was 16-fold lower than the black soil of Northeast China [52] and 9-fold lower than the results of Luo et al. [19], although a higher individual abundance of nematodes was obtained in this study. The main reason for this is not only because of differences between the nematode communities in the two regions, but differences in the methods used to estimate the nematode biomass [19,52]. Previous studies overestimated the nematode individual biomass because they used the adult body mass taken from the Nemaplex website to estimate the whole community, although more than 50% of individuals were juveniles [52]. Due to the lack of a feasible method for weighing the actual mass of nematodes, we used over 7000 individuals to measure the nematode length and width, and calculated biomass using an allometric growth equation. This method has been widely used to estimate nematode biomass [18,20,38] and has been shown to be accurate in providing useful information when comparing relative differences across different systems.

## 5. Conclusions

Compared with natural forests, intensively managed spruce plantations increased the daily carbon budget of total nematodes by approximately 52% through the herbivore channel in the 0–5 cm soil layer, while the daily carbon budget and production of herbivorous nematodes was decreased by approximately 60% in the moss layer. The nematode carbon production and budget are significantly promoted by individual abundance, while genus richness inhibited this in the subalpine forests. Environmental factors directly promoted the nematode carbon budget and production of bacterivores, fungivores, omnivores, and predators, while the carbon budget of herbivore nematodes was indirectly promoted by environmental factors through the omnivore top-down interaction. In conclusion, the planted ecosystems have a certain capacity to maintain the abundance, richness, and carbon budget of soil nematodes but increased the risk of herbivorous pests.

**Supplementary Materials:** The following supporting information can be downloaded at: https://www.mdpi.com/article/10.3390/f13030462/s1; Figure S1: an initial model for nematode carbon used in production and budget through food web bottom-up (A) and top-down (B) relationships; Figure S2: vertical distribution of relative daily carbon used in production (A) and daily carbon budget (B) of nematodes in subalpine plantation and natural forests; Figure S3: incremental percentage (%) of plantations than natural forests. DP: daily carbon used in production; DB: daily carbon budget; Total: total nematodes; He: herbivore nematodes; S: 0–5-cm soil layer; S: moss layer; Table S1: details of experimental areas in subalpine spruce forests; Table S2: average nematode individual density (ind./100 g dry matter) in different layers in subalpine spruce plantation and natural forests.

**Author Contributions:** Conceptualization, K.P.; methodology, H.Z. and X.S.; software, H.Z.; formal analysis, H.Z. and X.S.; investigation, X.W., L.Z., M.Z., T.T. and R.Z.; writing—original draft preparation, H.Z.; writing—review and editing, K.P., X.S. and B.A.; visualization, H.Z.; supervision, K.P.; project administration, K.P.; funding acquisition, K.P. and P.G. All authors have read and agreed to the published version of the manuscript.

**Funding:** This research was funded by the National Natural Science Foundation of China (31961133012), the National Science Centre of Poland (2018/30/Q/NZ9/00378), and the Second Tibetan Plateau Scientific Expedition and Research Program (2019QZKK0303).

**Institutional Review Board Statement:** Not applicable.

**Informed Consent Statement:** Not applicable.

**Data Availability Statement:** Not applicable.

**Conflicts of Interest:** The authors declare no conflict of interest.

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
