# Peer review of "Environmental Factors Indirectly Impact the Nematode Carbon Budget of Subalpine Spruce Forests"

_forests, doi:10.3390/f13030462_

Round 1

Reviewer 1 Report

This manuscript reports on the effects of environmental factors on nematode carbon budgets and other community metrics in planted vs natural forests. The topic is relevant and interesting. Also, appropriate methods were used. It is also worth noting that substantial efforts were made to generate the data. For example, identifying, counting and measuring such a large number (>7 000) of nematodes is no trivial task.

In general, the quality of the MS is of good quality. However, the MS requires substantial language editing. Some statements are unclear, while other sentences are poorly constructed. However, this can be fixed by asking a native speaker to correct the mistakes.

I made a number of recommendations in the attached pdf file. Mostly, these relate to clarifying statements. I also made some language suggestions on the first and second pages.

Reviewer 2 Report

The study was well-conducted, avoiding pseudoreplication, with adequate sampling procedure and statistical treatment. Results were straightforward, showing that the biomass of herbivorous nematodes was favoured by spruce plantation and that this biomass was lower in the moss layer compared to the underlying topsoil. However, the study is flawed by the conclusion that “planted ecosystems are valuable habitats for soil nematodes”. Clearly this concerns only root-feeding nematodes, which are in fact considered as pests for trees. The fact that moss carpets are impoverished in root-feeding nematodes just indicates that they are poor in tree roots, which is not novel. It looks like if results were already given by the authorities in charge of forest management. The increased burden of root feeders following the replacement of natural forests by planted forests cannot be considered as good for the forest ecosystem. It is surprising to see that the authors, which are nematologists (they were able to identify specimens at genus level), have no knowledge of the indices currently used by nematologists to assess the good health of soil through the analysis of nematode communities. See for instance https://link.springer.com/article/10.1023/A:1004571900425. As such, this paper cannot be published because it might be conducive to erroneous decisions in terms of forest management and land-use change. Another defect is the English language, which must be improved to become acceptable in an international journal.

Reviewer 3 Report

Dear editor,

I appreciate the opportunity to revise this interesting manuscript (forest-1589742). I think that this ms could be published in Forests MDPI after few improvements.

I recommend minor revisions.

Below, I have described some points that need adjustment:

General comment (GM) #1. Some points into the introduction and discussion are too vague. Authors need to use data/numbers to support their scientific statements.

                For example: L40-41; L42-44; L48-49; L61-64; L69-71; L75-78; L324-325; L327-328; L333=335; L337-340; L344-345; L351-354; L361-364; L368-372.

GM #2. Since the authors have described the soil nematode community using their function, I recommend describing the role of bacterivores, fungivores, omnivores, predators, and herbivores on each specific C compartment on C cycling. It would be interesting as a complement on L52-71.

GM #3. I would like to see a climograph (to illustrate abiotic variation, e.g., monthly mean temperature and accumulated precipitation on subsection 2.1), and a scheme (to illustrate the experimental design on subsection 2.2).

GM#4. A NMDS approach will show how dissimilar the nematode community was as influenced by the forest types and soil layers.

GM#5. Improve the statement about the first hypothesis. I would like to read more content about both niche complementarity and functional redundancy theories. Both theories are very close to the main results observed into this manuscript. The authors must describe it deeper in their discussion.

Round 2

Reviewer 2 Report

All changes I suggested on a previous draft of this paper were taken into account by the authors to improve it. In particular the final conclusion of the study has been changed, which is fully acknowledged.